# Fabrication of Energetic Composites with 91% Solid Content by 3D Direct Writing

**DOI:** 10.3390/mi12101160

**Published:** 2021-09-27

**Authors:** Yucheng Deng, Xinzhou Wu, Peng Deng, Fayang Guan, Hui Ren

**Affiliations:** State Key Laboratory of Explosion Science and Technology, Beijing Institute of Technology, Beijing 100081, China; 3120190177@bit.edu.cn (Y.D.); 3120200256@bit.edu.cn (X.W.); 3120195115@bit.edu.cn (P.D.); 3120180196@bit.edu.cn (F.G.)

**Keywords:** direct write ink, energetic composite, high solid content, combustion performance

## Abstract

Direct writing is a rapidly developing manufacturing technology that is convenient, adaptable and automated. It has been used in energetic composites to manufacture complex structures, improve product safety, and reduce waste. This work is aimed at probing the formability properties and combustion performances of aluminum/ammonium perchlorate with a high solid content for direct writing fabrication. Four kinds of samples with different solid content were successfully printed by adjusting printing parameters and inks formulas with excellent rheological behavior and combustion properties. A high solid content of 91% was manufactured and facile processed into complex structures. Micromorphology, rheology, density, burning rate, heat of combustion and combustion performance were evaluated to characterized four kinds of samples. As the solid content increases, the density, burning rate and heat of combustion are greatly enhanced. Based on 3D direct writing technology, complex energetic 3D structures with 91% solid content are shaped easier and more flexibly than in traditional manufacturing process, which provides a novel way for the manufacture of complicated structures of energetic components.

## 1. Introduction

Energetic materials present a high energy density and rapid energy release properties, and are sensitive to energy stimulation [1]. Additionally, it is precisely because of these characteristics that the manufacture of energetic materials has always been an attraction for scientific researchers. Traditional manufacturing processes, such as pressing, pouring and casting, exert force on an energetic system in terms of uniformity and large density, and these processes have a long manufacturing cycle and high safety requirements for operators, workshops and equipment [2,3]. For energetic materials, it is essential to increase the solid content in a formula to a high energy level under the prerequisite of fine formability. However, lowering the proportion of binder increases the difficulty of fabrication [4,5], as well as the potential hazards of heat accumulation in the processes of friction, extrusion, accumulation, and the spread between solid particles. Corresponding to reduced material manufacturing, additive technology can achieve digital precision manufacturing and realize human-machine isolation, with excellent application prospects in the processing and manufacturing of high solid content energetic materials.

Technology related to material additive manufacturing, also called 3D printing technology, is a rapid prototyping technology based on 3D CAD model data that are shaped by adding materials layer by layer. Three-dimensional printing technology has unique advantages over traditional technologies, such as its controllable and complicated structure, complicated geometry, high degree of automation, and safety processing [6,7]. Therefore, additive manufacturing has been widely studied and applied in biological [8,9,10], ceramics [11,12,13,14], metal forming [15,16], and concrete [17,18,19] spheres, as well as in other fields [20]. According to the standards set by the Subcommittee of materials and Testing Association of America (ASTM), additive manufacturing technology is divided into seven categories [21,22]. Among them, material spray forming technology [23], material extrusion technology [24], and photopolymerization curing technology [25] have been applied in the forming and manufacturing of energetic materials [23,25,26,27,28,29,30,31,32]. Material spray forming technology requires a material to have a lower viscosity so that it can be sprayed from small nozzles; however, photopolymerization curing technology requires more photosensitive resin in the molding material, and the solid content of both should not be high. As a result, using additive technology to realize the fabrication of energetic materials with a high solid content is a challenging research topic. As a kind of extrusion technology, direct writing technology has attracted particular attention owing to its advantages regarding low cost, high efficiency, relatively simplicity, and convenience [33,34,35]. Compared with the other two technologies, direct writing technology is more suitable for obtaining samples with a high solid content. Ye et al. [32] printed an CL-20 structure with a solid content of 56.25% using direct writing technology. The printed and molded samples present a 3D-honeycombed shape, the ink presents a high porosity, and the critical detonation size is very small, less than 60mm. Wang et al. [28] developed a nanothermite formulation with a solid content of 90%, and used the direct writing method for printing. The linear burning rate and flame temperature can be adjusted by changing the ratio of fuel to oxidant. Shen et al. [36] printed 90% solid content aluminum copper oxide thermite by direct writing technology. The binder is a mixture of multiple polymers. The energy flux can be easily adjusted by changing the equivalent ratio.

In this work, we have chosen Al/AP/HTPB composite energetic materials [37], which are widely studied and applied in the field of energetic materials. As a thermosetting formula based on solvent volatilization, it can control the solvent content in ink to improve safety. The binder content was changed to prepare various slurries with solid content of 80, 85, 90, and 91%. The rheology of slurries was studied; these properties are critical in terms of whether a slurry can be extruded from a nozzle. The micromorphology of samples with different high solid content was observed and the density was measured for compactness. The combustion performance of the samples was compared. Numerous complex structure samples were fabricated by direct writing technology, yielding a solid content of 91%.

## 2. Experimental Section

### 2.1. Materials

Aluminum and ammonium perchlorate composite powder (Al/AP) with an average diameter of 15 μm was obtained from Sichuan Hongbo Co., Ltd. (Mianyang, China). Hydroxyl-terminated polybutadiene (HTPB) with a number average molecular weight *Mn* = 3000 g/mol was purchased from the Liming Research Institute of Chemical Industry (Luoyang, China). Toluene diisocyanate (TDI, >98.0% (GC)) was obtained from Shanghai Macklin Biochemical Co., Ltd. (Shanghai, China). Ethyl acetate (AR grade) was bought from Tianjin Weisi Chemical Reagent Co., Ltd. (Tianjin, China).

### 2.2. Ink Preparation and Curing

First, we took a solid content of 80%. For example, 2.36 g HTPB was dissolved in 2.5 g ethyl acetate. After the HTPB was completely dissolved, 10 g of Al/AP composite powder was added into the mixture solution with ultrasonic oscillation for 20 min. Next, 0.14 g TDI was added to the solvents and stirred for 4 h to distributed components uniformly. The ink suitable for preparation was obtained until the ethyl acetate content in the mixture was 0.08 g/g; then, ink was loaded in the container for 3D printing. The solid content and the specific HTPB and TDI dosages are shown in Table 1.

Regarding the 95% solid content, although we also prepared the corresponding slurry sample, on the one hand, the viscosity of the slurry was large and difficult to extrude; on the other hand, the solvent in the slurry was more volatile and easier to print during the printing process. The nozzle was blocked, so the 95% solid content samples were not tested for printing and characterization afterwards.

According to the ratio of HTPB and TDI, the hydroxyl group in HTPB and isocyanate group in TDI will react with each other to form a polyurethane network [38]. This reaction will last for 4–5 days at about 50 °C, so the whole system can be firmly bonded together [39].

### 2.3. Characterization and Testing

To observe the micro characteristics of the samples, a Hitachi S-4800 (HITACHI, Tokyo, Japan), scanning electron microscope (SEM) was used with a maximum resolution of 2.0 nm. An energy dispersive spectrometer (EDS) was coupled to the scanning electron microscope (SEM) to analyze the type and content of component elements in the material. The combustion rate was measured by a FASTCAM APX RS high speed camera. The combustion heat of the sample was measured with an automatic calorimeter (TRHW-7000C, Hebi, China). The rheological properties of the ink were tested by an MCR 102 rotary rheometer.

### 2.4. Equipment and Process Adjustment

The 3D direct writing extrusion system used in the study was Nordson E5(NORDSON, Westlake, OH, USA). It was divided into four parts; motion platform, computer, dispensing machine and pressure source. First, the modeling path was predicted in advance, and the relevant instruction programming language was input into the software of the computer. Additionally, the motion platform received the command and controlled the nozzle to move on the X, Y, and Z axes. The dispensing machine used external N_2_ as the pressure source, switched the gas valve and controlled the pressure of the slurry squeezing. The schematic diagram is shown in Figure 1.

The rheological properties of each component are different for various dispersion performances. In the printing process, the process parameters suitable for each component were found, as shown in Table 2. The application of these process parameters means the slurry can be printed continuously and smoothly, presenting a better effect.

## 3. Results

### 3.1. Rheology

The sample preparation flow chart is shown in Figure 2. Taking an 85% solid content formula as an example, an ordinary stainless needle was selected for printing, and the diameter of the needle was 1.3 mm. This was then externally connected to a high-purity nitrogen gas cylinder, and the output pressure of the air pump was adjusted to 30~50 psi. We then controlled the motion device, selected the slow speed gear, and set the printing rate as 1~2 mm/s.

Slurry printed by direct writing technology requires suitable rheological properties, and the viscosity at different shear rates is important for the slurry to be extruded smoothly. At low shear rates, the slurry needs to have a high viscosity so that it does not drop from the needle under the influence of gravity. On the contrary, at high shear rate, the slurry needs to have a lower viscosity so that it can be extruded smoothly.

The rheological properties of four kinds of solid content ink are shown in Figure 3. All four sample inks exhibited shear thinning characteristics [40,41]. It can be seen from the figure that the shear viscosity of the four samples decreases with an increase in shear rate, and the viscosity drops from hundreds to tens. At a low shear rate area, viscosity increases with a high solid content and fewer adhesives and solvents; on the contrary, under a high shear rate area, the higher the solid content, the lower the viscosity.

Figure 4 shows the elastic modulus and viscous modulus diagrams of these solid contents. It can be seen that, when the solid content is 80%, the elastic modulus is always smaller than the viscous modulus. At this time, the slurry exhibits more fluid viscous properties, which is also reflected in the sample collapse in subsequent printed samples. The slurry cannot maintain the shape and appearance of the nozzle just after the nozzle is extruded. In addition to the solid content of 80%, the data of solid contents of 85, 90, and 91% show that the prepared printing paste is not a pure fluid or solid, but instead a semi-solid with different properties under different strains and shear rates. The slurry exhibited solid elastic properties in areas with low shear rate, and then exhibited fluid viscous properties in areas with high shear rate. Therefore, this enables the high viscosity slurry to have sufficient fluidity to extrude through a small nozzle and then return to the previous high viscosity status. As such, it retains proper fluidity and processability during printing, and has no significant influence on the final shape; this characteristic is ideal for 3D printing.

### 3.2. Micromorphology

The SEM images of four samples’ surfaces and cross-sections with different solid contents obtained using the direct writing technique are shown in Figure 5.

The surface of the sample was observed by scanning electron microscopy, and the dust and impurities that may adhere during the printing process and the transfer process were ignored. From Figure 5, due to the decrease in binder content, the higher the solid content, the coarser the surface. From EDS, it can be seen that Al, AP and binder are closely combined, Al and AP are surrounded by the binder, and the distribution of each component is relatively uniform.

Although the four samples with different solid contents have the same solvent content when printing, due to the different binder contents, the binder fills the pores left after solvent volatilization. There are no obvious pores on the surface or inside of the samples with solid contents of 80 and 85%, while there are some unevenly distributed pores on the surface and inside of the samples with solid contents of 90 and 91%. After solvent volatilization, the binder content was not enough to completely fill the pores.

When the solid content rise from 80 to 91%, the contact angle between the sample and the substrate reduce from 144° and 143° to 48° and 50°, and the height of the sample increase from 0.55 to 0.9mm. With the increase in solid content, the collapse of the sample is less obvious, and the molding effect is better and closer to the shape when it is only extruded from the needle. At the same time, combined with the surface SEM images and element analysis images, it can be seen that the components in the samples were evenly distributed; the spherical particles were Al particles, the bulk materials were AP, and the binder components HTPB and TDI were between the particles.

### 3.3. Density and Combustion Performance

At room temperature, an external heat source was used for ignition, and the combustion images of four groups of samples were captured under the same experimental conditions through high-speed photography. After processing the picture, the burning speed is equal to the flame stroke divided by the time. The combustion images are shown in Figure 6.

For energetic materials, density is an important factor affecting burning rate, energy, heat of combustion, etc. The density of 3D printed samples is shown in Figure 6. With the rise in solid content, the density increases from 1.587 to 1.700 mm/s.

Many factors affect the burning rate [42], such as density, width, etc. Herein, we analyzed the effect of solid content on the burning rate of the samples. Figure 6 shows the combustion process of the four samples. The four samples with different solid contents can burn steadily and continuously, which indicates that the printed samples have good continuity. There is no intermittent combustion phenomenon or a sudden darkening of the flame, indicating that the components of the slurry prepared by this method are uniformly mixed. With the increase in solid content, the combustion rate of the sample increases gradually from 1.34 to 3.65 g·cm^−3^. The higher the solid content is, the more Al/AP and less binder are found in the samples. Combined with the micro morphology of the four samples, it can be seen that there are some tiny pores left by solvent volatilization in the samples with a high solid content, which enhances the convective combustion of the flame during combustion, and makes the flame reach the non-ignited area faster; this also shows an increase in the burning rate. From the images in Figure 6, it is clear that the combustion process of the whole system gradually becomes intense, emitting more light and brightness, and the size of flame is also larger.

The heat of combustion of an energetic material component is the heat released when it is fully burned in an oxygen environment, which can indicate its energy performance. Table 3 shows the density, burning rate and heat of combustion data of four different solid content samples. It can be seen from Table 3 and Figure 6 that the samples printed by a 3D device can burn continuously, and the heat of combustion increases with the increase in solid content during this process from 39.761 to 40.636 kJ·cm^−3^.

### 3.4. Special Shaped Structure Printing

In order to verify the performance of 3D printing in the process of a three-dimensional structure and the completion of complex structures, several complex shapes of three-dimensional graphics were printed by 3D printing (Figure 7). The print videos are shown in Appendix A. The samples can be printed smoothly and stably from the nozzle in the printing process, and the printed samples can maintain the printing shape without any auxiliary devices (a substrate was placed underneath when printing). Three-dimensional printing is feasible and promising in printing complex structures.

## 4. Conclusions

We designed and printed four kinds of Al/AP/HTPB ternary energetic composites with a high solid content by adjusting various process parameters of 3D direct writing equipment. The microstructure, density, and combustion performance were studied by adjusting the solid content. These four kinds of solid content inks can be used for direct writing, which proved that the inks have suitable rheological properties. As the solid content increases, both the density and the heat of combustion increase, the burning rate becomes faster, and the combustion becomes more intense. More importantly, samples with optional complex structures can be manufactured to control combustion performance. It is a feasible and promising method to print composite solid propellant by direct writing technology, which can easily and quickly produce a variety of complex structures, and is expected to be applied and industrially produced.

## Figures and Tables

**Figure 1 micromachines-12-01160-f001:**
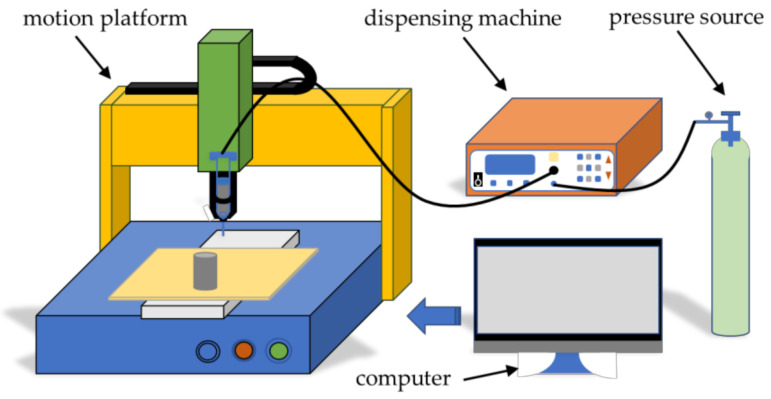
A brief diagram of a direct writing device for preparing energetic composites.

**Figure 2 micromachines-12-01160-f002:**
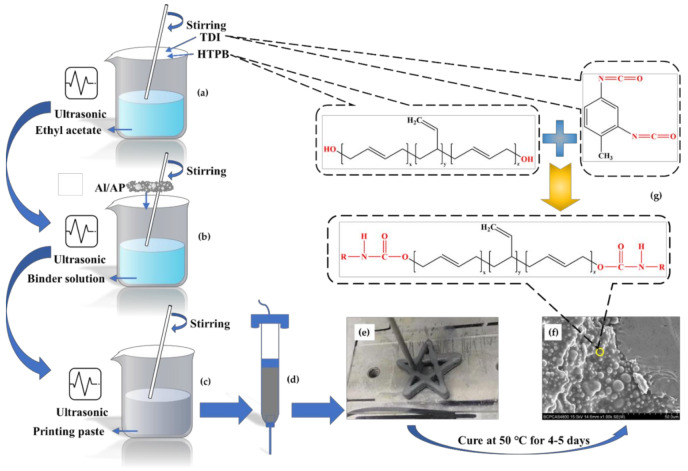
Schematic diagram of slurry preparation and curing mechanism of 3D printing. (**a**) TDI and HTPB were added to ethyl acetate; (**b**) Al/AP was added to the binder solution; (**c**) stir all components for 4 hours; (**d**) transfer the stirred slurry to the syringe; (**e**) install the syringe on the 3D printer and start printing; (**f**) after curing, the internal components are closely connected; (**g**) reaction mechanism of TDI and HTPB.

**Figure 3 micromachines-12-01160-f003:**
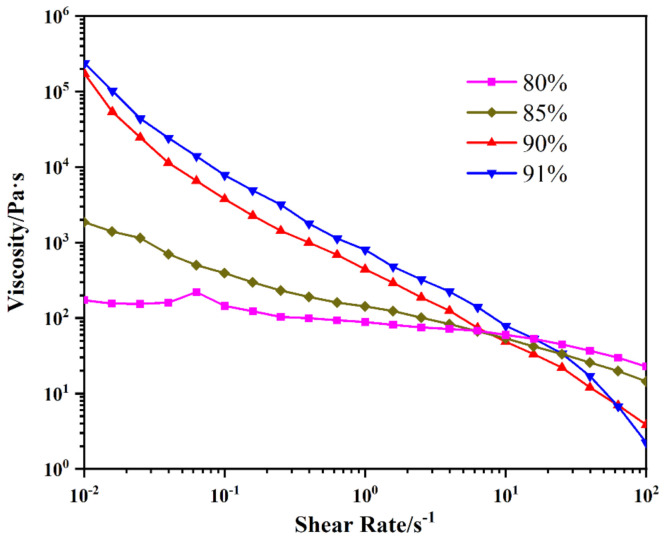
Rheological properties of slurry with different solid content.

**Figure 4 micromachines-12-01160-f004:**
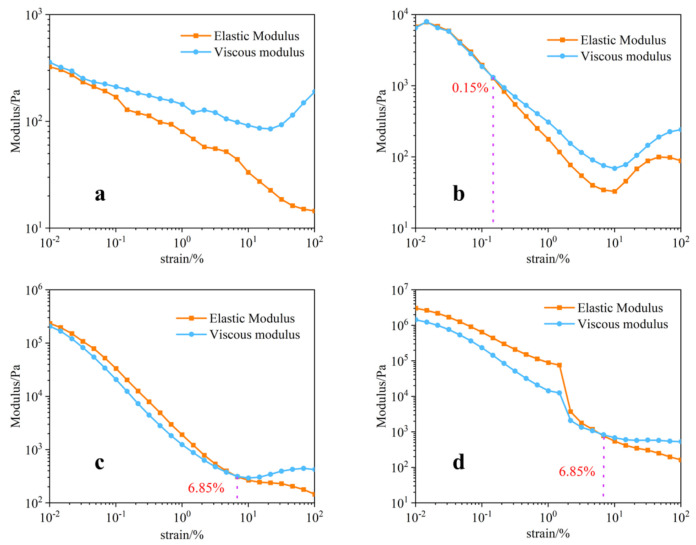
Elastic modulus and viscous modulus images of four samples with different solid contents: (**a**) image of 80% sample slurry; (**b**) image of 85% sample slurry; (**c**) image of 90% sample slurry; (**d**) image of 91% sample slurry.

**Figure 5 micromachines-12-01160-f005:**
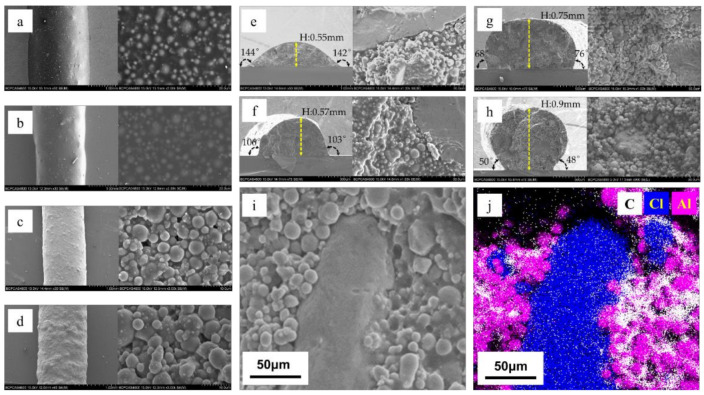
SEM images and element analysis image of four samples with different solid contents: (**a**) SEM images of 80% sample surface; (**b**) SEM images of 85% sample surface; (**c**) SEM images of 90% sample surface; (**d**) SEM images of 91% sample surface; (**e**) SEM image of 80% sample cross section; (**f**) SEM image of 85% sample cross section; (**g**) SEM image of 90% sample cross section; (**h**) SEM image of 91% sample cross section; (**i**) SEM image of 91% sample surface; (**j**) element analysis image of 91% sample surface (white is C, blue is Cl, and pink is Al).

**Figure 6 micromachines-12-01160-f006:**
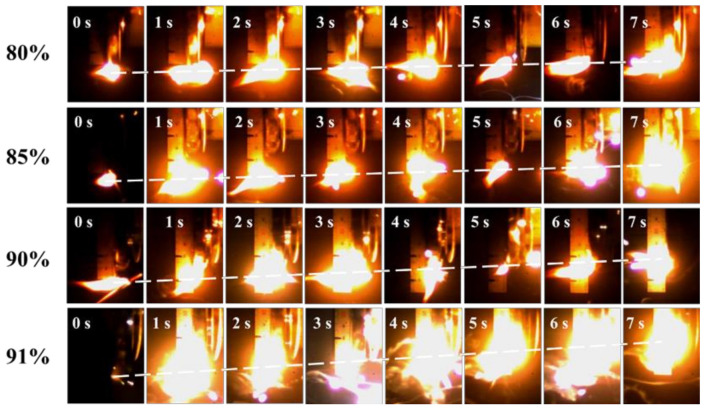
Combustion images of samples with different solid contents.

**Figure 7 micromachines-12-01160-f007:**
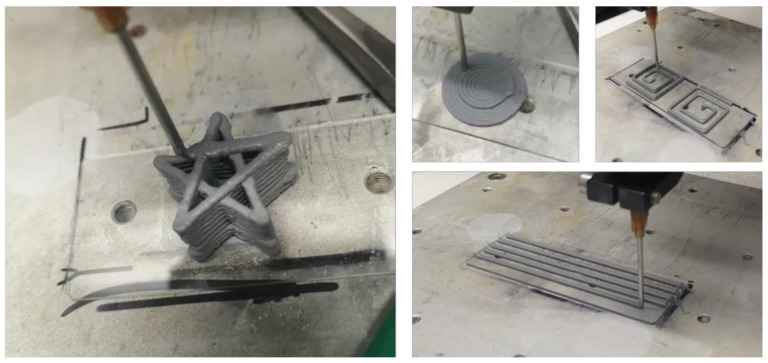
Several complex structures through 3D printing.

**Table 1 micromachines-12-01160-t001:** The solid content and the specific HTPB and TDI dosage.

Solid Content/%	Solid Content/%		Binder Content/%	
	Al/g	AP/g	HTPB/g	TDI/g
80	7.5	2.5	2.36	0.14
85	7.5	2.5	1.66	0.10
90	7.5	2.5	1.05	0.06
91	7.5	2.5	0.93	0.05
95	7.5	2.5	0.50	0.03

**Table 2 micromachines-12-01160-t002:** Process parameters of components in printing process.

Solid Content/%	Needle Diameter/mm	Syringe Pressure/psi	Printing Speed/mm·s^−1^
80	1.3	30	2
85		36	1.5
90		43	1
91		45	1

**Table 3 micromachines-12-01160-t003:** Density, burning rate and heat of combustion of samples with different solid content.

Solid Content/%	Density/g·cm^−3^	Burning Rate /mm·s^−1^	Heat of Combustion/kJ·cm^−3^
80	1.587 ± 0.04	1.34 ± 0.03	39.761 ± 0.09
85	1.636 ± 0.03	1.55 ± 0.01	40.099 ± 0.11
90	1.689 ± 0.02	2.22 ± 0.02	40.424 ± 0.13
91	1.700 ± 0.05	3.65 ± 0.03	40.636 ± 0.10

## Data Availability

Data are contained within the article or Appendix A. The data presented in this study are available in the Appendix A.

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
