# Peer review of "Fabrication of Energetic Composites with 91% Solid Content by 3D Direct Writing"

_micromachines, 2021, doi:10.3390/mi12101160_

Round 1
Reviewer 1 Report
The contribution from Yucheng Deng and coauthors titled “Fabrication of energetic composites with 91% solid content by 2 3D Direct Writing” contains a demonstration of 3D printing of solid propellant. This manuscript has an interesting premise but requires major revisions in many areas.
- AL/AP should be written out fully in abstract
- Phrase “certain mechanical properties” in abstract should be revised
- Phase “pressure to squeeze” in abstract should be revised
- The first sentence of the introduction talks about energetic materials but requires references to establish a literature precedent
- Line 41 additional should be changed to additive
- Line 47 “As a result, using additive tech-47 nology to realize the fabrication of energetic materials with high solid content is a chal-48 lenging and novel research topic.” Up to this point the authors have explained additive manufacturing but not defined energetic materials specifically or explained why those materials would be challenging or novel. Recommend rephrasing or restructuring.
- The term “solid propellant” should be used and explained in the introduction, not just the conclusion
- Line 63 “in this work” is repetitive and should be removed
- Authors should mention if 95% solids were tested and just didn’t work well
- Line 85 has sentence fragment, consider revising
- Phrasing on line 133 “bigger and bigger” should be revised
- The rheological discussion starting on line 131 needs significant revision to explain the rheological behavior at a technical level. The sentence on 136-137 is of questionable accuracy. Some statement should be made about the 91% content rheological behavior vs the lower solids content inks.
- Line 146-147 has some major grammatical issues including punctuation issues and a sentence fragment that significantly detracts from clarity
- In storage/loss modulus discussion beginning on line 143, the term “turning point” is used repeatedly, which is not a term that has technical meaning and should be revised to a correct term. What made the 91% content preferred over the 90% content ink?
- Line 167 is the term surface finish referring to roughening? The phrase “worse and worse” needs to be reconsidered so-as to describe quantitatively what is happening in a technical way
- From the SEM/EDS image in Figure 5j it is not clear that the Al is distributed throughout. Consider a supporting figure to show this more clearly
- I couldn’t find Fig S1 referenced in the text
- Line 206 “tiny pores” should be quantified to some extent
- Consider making Figure 2d a couple 2D plots
- Angles in chemical structures in Figure 2 need to be corrected
- Conclusions should be directed towards the combustion capabilities of the materials
Reviewer 2 Report
The manuscript by these authors is an interesting contribution about the setting of printing parameters for obtaining particular structured materials. The work was carried out by using various techniques and the results are worthy of publication. Anyway, some aspects need attention. The English must be revised, the introduction must be completed by adding relevant references, the conclusions must be improved by reporting all the results they obtained in the work. Other suggestions and typos to be corrected are highlighted in the attached .pdf.

Round 2
Reviewer 1 Report
The authors have provided detailed feedback responding to many of the issues that needed to be addressed with the manuscript. Grammatical, phrasing, and terminology issues continue to be very frequent in the manuscript, but in many areas the discussion has been noticeably improved. I have highlighted numerous areas in the text pdf where the authors should still examine for grammatical, phrasing, or clarity issues.
The introduction has been improved and the formulation of energetic materials is now specified in the abstract, but the authors still need to mention the specific materials they will be using within their introduction. For instance, the authors make the general statement “In this work, a thermosetting solid propellant formula based on solvent volatilization was studied, which can control the solvent content in ink to improve the safety.” The authors would be better served by referring to the literature at this point in the text regarding specific energetic material formulations, their relative performance, and also include specific information about their formulation. Why was this formulation chosen specifically out of other energetic materials?
The authors state that “Compared with the other two technologies, direct writing technology is more suitable for obtaining samples with high solid content.” Are there literature references for solid contents achieved with DIW? Has anyone else extruded energetic materials similar to this to-date? If so, mention the state of the field as it relates and if there are references relevant to this question, include them.
The discussion of the 95% solid content could likely be moved out of the introduction and into the methods as well as the discussion. For example, in methods, include 95 percent as one of the formulations, and then in discussion include the reasoning for not studying this formulation further.
The statement in the introduction saying that rheology was probed preliminarily should be rephrased, since preliminary studies by their nature are not ready to be published. Instead, the authors should briefly reference the assessment they were able to make of their materials directly from rheology and allude to why it was relevant to their characterization or conclusions.
Why were the complex structures that were printed selected by the authors? “Numerous complex structure samples were fabricated with solid content of 91% by direct writing technology.” Was there a scale or aspect ratio that was necessary to target with these structures?
Strongly recommend Figure 6 revised to 2D plot or a table. As-is it does not clearly show trends or give the reader a concise way to understand the data.
Standard deviations should be included for the combustion data in Table 3, especially because the heat of combustion reported for the different solids content is so close across the formulations. Were the heats of combustion statistically significant between formulations? Information on the methods used for combusting the materials should be detailed in the Methods. How did the authors determine burning rate? What techniques were used to ensure accurate measurement of heat of combustion?
The solids content of the inks clearly affects the density and therefore the energetics of the filaments. The discussion has been improved significantly throughout the paper to make these claims more clear. The authors describe how 3DP is primarily useful for the safety and processing concerns that it may help alleviate. Did the authors study the density resulting from the 3DP structure geometries themselves? Have the authors considered a study where they print rectilinear shapes with different infill and show how the energy outputs can be further controlled by controlling 3D geometry? Relating to this, could the authors compare the filament density for the different geometries they printed and comment on how the “arbitrary” aspect of the structures they are able to print is particularly important for energetic materials specifically. Currently the authors state, “More importantly, samples with arbitrarily complex structures can be manufactured to control combustion performance,” and some discussion about the density of the 3DP structures and how geometry affects combustion performance (in addition to the existing discussion about how filament solids content affects energetic performance) would be of interest.
I believe that the authors should complete another major revision of their manuscript, since the findings are of sufficient interest to the community once numerous substantial issues are addressed.

Round 3
Reviewer 1 Report
The authors have completed the changes requested and enhanced their data presentation and discussion throughout. I would recommend this work for publication.